# Application of Doehlert Experimental Design for Optimization of a New-Based Hydrophilic Interaction Solid-Phase Extraction of Phenolic Acids from Olive Oils

**DOI:** 10.3390/molecules28031073

**Published:** 2023-01-20

**Authors:** Bochra Bejaoui Kefi, Khaoula Nefzi, Sidrine Koumba, Naceur M’Hamdi, Patrick Martin

**Affiliations:** 1Laboratory of Useful Materials, National Institute of Research and Pysico-Chemical Analysis (INRAP), Technopark of Sidi Thabet, Ariana 2020, Tunisia; 2Department of Chemistry, Faculty of Sciences of Bizerte, University of Carthage, Bizerte 7021, Tunisia; 3LR11INRGREF0 Laboratory of Management and Valorization of Forest Resources, National Research Institute of Rural Engineering, Water and Forests (INRGREF), Carthage University, Ariana 2080, Tunisia; 4Unité Transformations & Agroressources, ULR7519, Université d’Artois-Uni LaSalle, F-62408 Bethune, France; 5Research Laboratory of Ecosystems & Aquatic Resources, National Agronomic Institute of Tunisia, Carthage University, 43 Avenue Charles Nicolle, Tunis 1082, Tunisia

**Keywords:** Doehlert experimental design, olive oil, phenolic compounds, solid-phase extraction

## Abstract

In this work, a rapid, precise, and cost-valuable method has been established to quantify phenolic compounds in olive oil using new-based hydrophilic interaction solid-phase extraction (SPE). Boehlert’s experimental design applied the determination of the optimal operating conditions. An investigation into the effects of the methanol composition (50–100%), the volume of eluent (1–12 mL), and pH (1–3) on the extraction of phenols acids and total phenols from Tunisian olive oils was performed. The results showed that the extraction conditions had a significant effect on the extraction efficiency. The experiment showed that the greatest conditions for the SPE of phenolic acids were the methanol composition at 90.3%, pH at 2.9, and volume at 7.5 mL, respectively. The optimal conditions were applied to different types of olive oils, and it could be concluded that larger concentrations of polyphenols were found in extra virgin olive oil (89.15–218), whereas the lowest levels of these compounds (66.8 and 5.1) were found in cold-pressed crude olive oil and olive pomace oil, respectively.

## 1. Introduction

In recent years, plant polyphenols have drawn increasing attention due to their potent antioxidant properties. They represent a large and diverse group of substances abundantly present in most fruits, herbs, and vegetables [1]. Phenolics are secondary metabolites that are naturally present in several different plants and fruits [1,2], as well as in olives and olive oils. Polyphenols provide an antioxidant capacity, which has many health effects and are expressed in the form of inhibiting the oxidative process by scavenging free radicals and reactive oxygen species, which can help prevent oxidative stress [2]. Many researchers had developed new methods for the identification and extraction of these polyphenols from natural and medicinal plants and agro-food products [3,4,5]. Vegetable oils belong to this category of food products widely used in our daily diet. Their evaluation has progressed a lot through physical–chemical methods of analysis. They have contributed to the determination of their chemical composition and their organoleptic and therapeutic qualities [6,7]. According to the International Olive Oil Council (IOC), the main producers of olive oil in the European Union EU are Spain, Italy, Greece, and Portugal, with a total production of 2247 tons. Outside the EU are Turkey (193 tons), Morocco (200 tons), Tunisia (140 tons), and Syria (104 tons). Olive oil is therefore consumed in large quantities in the Mediterranean Basin and is one of the main reasons for the benefits of the Mediterranean diet [8,9]. The composition of olive oil is complex, variable, and varied. It is made up of major compounds (97 to 99%, triglycerides) that are a source of energy and minor compounds that are a nutritional source (1 to 3%; examples, fatty acids, phospholipids, sterols, tocopherols, and phenolic compounds) [10,11,12,13]. Several green nonconventional methods have been developed for reducing the operational time and usage of organic solvents, such as ultrasound-assisted extraction, microwave-assisted extraction, enzyme-assisted extraction, pressurized liquid extraction, supercritical fluid extraction, high hydrostatic pressure extraction, pulsed electric field extraction, and high voltage electrical discharge extraction [14,15,16,17].

These compounds are thermolabile [18], so they require derivatization before analysis [19,20,21,22,23]. This additional treatment generally leads to stability problems for these compounds. This does not prevent the coupling of GPC to mass spectrometry (MS) to identify the isomers of phenolic acids using the fragmentation patterns provided [21]. However, most analyses of phenolic acids are based on liquid chromatography [24,25], generally used with UV [26,27,28]. Still, the detectors associated with this technique do not allow direct analyses of these samples. An extraction and preconcentration step before the analysis is therefore necessary. This treatment must be adapted to the properties and complexity of the matrix and must consider the physicochemical properties of the compounds to be analyzed: polarity, solubility, acid–base character, chirality, and thermosensitivity. Several methods of extraction of phenolic acids have been developed, namely, acid hydrolysis, saponification, enzymatic reactions, liquid–liquid extraction, microwave-assisted extraction, pressurized fluid extraction, supercritical fluid extraction, and solid-phase extraction [29,30,31,32,33]. Indeed, researchers are increasingly using methods that are always reliable in terms of yield and efficient in terms of time and cost and that limit the consumption of solvents [29,32,33,34]. These experiment matrices allow us to estimate the main effect of each influencing factor and the interactions between the factors.

Solid-phase extraction meets all these requirements, and thanks also to the multitude of stationary phases used, it is more and more coveted. The reverse-phase column mechanism is the most widely used. As for the hydrophilic interaction mode, only a minority of researchers have used it for the separation of these acids [29,35,36,37,38]. Therefore, an extraction method is reliable if it keeps the analyte intact and allows the recovery of all of it during the process. This requires the optimization of the factors influencing the extraction process [35,37]. In most cases, the mono-factorial approach is adopted, varying only one factor at a time while keeping the others constant [29]. However, the major drawback of this method is that it does not effectively describe the effects of the parameters and the interactions between them on the extraction process and the response [29,39]. Another disadvantage of classical optimization is sometimes the high number of experiments required for the optimization. In recent years, the design of the experiment’s method has been applied since it provides a rigorous approach to problem-solving. The principle of the method is not to study all the points of the mesh, but only certain points were chosen for their particularity of orthogonality. This experimental design allows us not only to study many factors and to know their influences but also to acquire information on the possible interactions between them. It will allow a quick and unequivocal interpretation of the test results by providing the best possible accuracy of the results and the modeling of the studied system [29,39,40]. In this context, this study aims to simultaneously optimize the extraction of phenolic acids and develop a new and simple solid-phase extraction method based on the hydrophilic interaction of phenolic acids from olive oils. A two-level full factorial design was used to estimate the experimental variables, including which are X_1_: the percentage of the methanol, X_2_: the volume of the eluent solvent, and X_3_: the pH of the elution solvent.

## 2. Results and Discussions

### 2.1. Screening Approach

The main goal of this study was to optimize a simple and reliable method to be used for phenolic profiling a large set of olive samples. The parameters considered for the optimization are X_1_: the percentage of the methanol, X_2_: the volume of the eluent solvent, and X_3_: the pH of the elution solvent. The resulting design of all the combinations of the different levels assigned to the factors in the chosen experimental area, as well as the extraction yields obtained with the 13 extractions, are presented in Table 1. An analysis of the Pareto chart (Figure 1) showed that the pH of the eluent phase has the highest effect on the extraction yield of phenolic acids, followed by the composition in methanol. However, the volume of the elution solvent (methanol/water) did not influence the studied response in the considered experimental range. As shown in Figure 1, to increase the extraction yield of phenolic acids in the chosen experimental field, the pH of the eluent phase should be maintained at a high level, while to decrease the background signal, this factor had to be kept at a low level. The pH of the sample to be extracted played a very necessary role in the SPE procedure [29,41].

### 2.2. Application of Doehlert Design to Optimize Experimental Variables

#### 2.2.1. Experimental Design

The experimental responses allow the evaluation of the efficiency of extraction yield. They are closely related to the factors influencing this response (Table 1). We have chosen a Factorial matrix of the experiments. These experiment matrices allow us to estimate the main effect of each influencing factor and the interactions between the factors [42,43,44,45,46]. The design resulting from all the combinations of the different levels attributed to the factors in the chosen experimental domain, as well as the extraction yields obtained with the 13 extractions presented in Table 1. The parameters considered for the optimization of the extraction are the percentage of methanol (X_1_), the volume (X_2_), and pH (X_3_). The response is the retention efficiency expressed in terms of the extraction yield (%).

The obtained extraction yield for the experiments fluctuated between 35 and 73.5%, corresponding to the minimum and maximum values for runs 6 and 9, respectively (Table 1). The highest value of extraction yield (73.5%) agreed with the results reported in many studies [46,47]. The design results from all possible combinations of the different levels assigned to the factors in the chosen experimental domain of the Doehlert approach [29,42,45], as well as the extraction yields obtained during the 13 extractions expressed in %, are shown in Table 1 [47].

#### 2.2.2. Analysis of Significant Factors

The statistical significance of each term (linear, Interaction, and quadratic) is reported in Table 2, obtained from the analysis of variance. The results obtained show that the percentage of methanol, volume, and pH have a significant influence (*p* = 0.009; *p* = 0.001, and *p* = 0.009) on the retention yield. The pH (X_3_) played a primary role in improving the extraction yield. Our results followed those previously obtained by Živković et al. [45], Sharmila et al. [47], and Arruda et al. [48].

#### 2.2.3. Model Fitting and Statistical Analysis

The analysis of variances was performed to determine if the equation and the quadratic model are significant [43,49]. The results of the second-order model are shown in Table 2. The model F-value of 9.20 implies the model is significant. There is only a 4.71% chance that an F-value this large could occur due to noise. The results demonstrated that the model was significant, as evidenced by the high value of the F-test and a low *p*-value [50,51].

#### 2.2.4. Equation with Coded Factors

An equation with coded factors was used to predict the response variable for the given levels of each factor. The low and high levels are coded (−1) and (+1), respectively. The final quadratic equation obtained in terms of the actual factors is given below:Y = 72 + 10X_1_ + −9.3125X_2_ + 0.0625X_3_ − 3.375X_1_X_2_ − 0.375X_1_X_3_ + 0.25X_2_X_3_ − 9.25X_1_X_2_ − 11.125X_2_X_2_ − 9.625X_3_X_2_

The polynomial mathematical model developed for optimization is a second-degree model. Figure 2 shows the curves of the predicted vs. observed values to confirm the goodness of fit. For each coefficient in the regression model, the significance was assessed by the corresponding *p*-values [52]. The coefficient of determination (R^2^) of the regression models was 0.96, and the value of the predicted coefficients of determination was 0.92, signifying a better correlation between the response values and factors [53]. This suggests a high degree of connection between the observed and the predicted values (Figure 2). Consequently, the applied model is suitable for the prediction of the extraction yield in the range of the experimental variables [54,55,56].

#### 2.2.5. Effects of Interactions of the Different Factors on Extraction Yield

The response surface graphs obtained (Figure 3) facilitate the visualization of the main and interactive effects of the factors of variation on the response variable. The graphical analysis of this figure shows that the polarity of the mobile phase is a very important factor in the extraction of phenolic acids from the oil. Its effect is positive for the studied response. The pH is the most influential factor in the extraction yield. An increase in pH results in an increase in the extraction yield of phenolic acids. We find that the pH and methanol composition were the two factors that had more influence on the extraction of phenolic compounds. The interactions (X_1_X_2_, X_1_X_3_, and X_2_X_3_) and quadratic coefficients (X_1_X_2_, X_2_X_2_, and X_3_X_2_) of the model had significant values (*p* < 0.01). Similar results were described by Yahia et al. [42], Fratoddi et al. [52], and Maran et al. [55].

### 2.3. Determination of Optimal Conditions

Different factors can affect the SPE efficiency; therefore, their optimization through a multivariate approach is recommended, especially when these factors are correlated [46,47,48]. According to the results of the optimization studies, optimal conditions were chosen as optimal values for the extraction of phenolic acids from olive oils. The experiment showed that the best conditions for SPE were the methanol composition at 90.3%, pH at 2.9, and volume at 7.5 mL, respectively, with a Desirability of 0.703. Comparable results have been found in many studies [29,45,51].

### 2.4. Application on Tunisian Olive Oils Samples

The chosen operating conditions (percentage of methanol equal to 90.3%, the volume of methanol 7.5 mL, and pH of the eluting solution equal to 2.9) allow obtaining an extraction yield of phenolic acids of about 94.5%. The determined optimal conditions were applied for the identification and quantification of some phenolic acids in Tunisian olive oil samples. Five samples of olive oils were studied, and three types were chosen: extra virgin olive oil, olive pomace oil, and raw oil freshly cold extracted in the laboratory of INRAP Sidi Thabet, Tunisia. The concentrations of some acids identified in these different samples are calculated in Table 3.

The total phenolic acids are very high in the sample of extra virgin olive oil (89.15–218). This value is lower for cold-pressed crude olive oil (66.8). The total phenolic acids are very low for olive pomace oil (5.1). This variation in the total phenolic acids can be due to several factors, among which are the climate, the degree of maturity of the olives, the technological processes of elaboration of the oil, and the genetic or varietal factor that remains the most dominant.

## 3. Materials and Methods

### 3.1. Samples

Different virgin olive oil samples were taken from various locations in Tunisia. Three of them were commercial olive oils: Châal, Hikma, and Olivetta. The other samples were extracted from Cv. Chetoui olive fruits assembled from two different locations in the north of Tunisia (Bizerte and Kef), and Cv. Chemlali olive fruits were collected from the center (Monastir) and southern arid regions (Mednine). The olives were picked by hand, and only the undamaged, fresh, and healthy ones were selected. The olive oils were extracted with a laboratory mill (Abencor). The oil obtained was slowly mixed for 30 min at room temperature, centrifuged without the addition of chemicals at 3500 rpm for 4 min, then transferred to amber glass bottles and kept in the freezer at −20 °C until analysis.

### 3.2. Reagents

All chemicals and solvents used were of analytical grade. Ultrapure water was obtained using a Milli-Q SYSTEM 5 (Millipore, Elix). For HPLC analyses, methanol was from Normapur, and formic acid was from Fischer. HPLC-grade hexane and acetonitrile were purchased from Prolabo, Chromanorm, and Normapur, respectively. The phenolic acids standards (97–99% purity) were obtained from Fluka (ortho and para coumaric acids and p-hydroxybenzoic acid); Sigma-Aldrich (gallic acid, caffeic acid, vanillic acid, syringic acid, ferulic acid, gentisic acid, benzoic acid, and cinnamic acid); and Merck (salicylic acid). Individual stock solutions containing 1 g L^−1^ of phenolic acids were prepared in a methanol/water mixture (50/50, *v/v*). Various dilutions were performed with the same solvent mixture for calibration purposes and SPE optimization. Solutions were kept in amber glass bottles at 4 °C.

### 3.3. Instrumental Analysis

Phenolic acids were analyzed using a Hewlett Packard (HP) series 1100 chromatograph and detector. The HPLC apparatus was equipped with a degasser (G1322A), a quaternary pump (G1311A), a column thermostat (G1316A), and an autosampler (G1313A). Detection was achieved at 280 nm with diode array detector (DAD) type G1315A. The stationary phase was a C18 Hypersyl ODS column (250 × 4 mm, i.d.) with a particle size of 5 µm and thermostated at 20 °C. The mobile phase consisted of two solvents, A: water/formic acid (99/1, *v/v*) and B: methanol/formic acid (99/1, *v/v*). Gradient elution was applied at a flow rate of 1 mL min^−1^. Solvent B was gradually eluted from 10 to 100% in 25 min. The sample volume injection was 20 µL, and the UV absorbance was determined at 250, 280, 300, and 320 nm.

### 3.4. Solid-Phase Extraction Procedure (SPE)

The SPE method based on hydrophilic interaction was done using MgO-SiOH (15:85; high purity; particle size ranged between 150 and 250 µm)-bonded silica-phase cartridge (6 mL, 1000 mg, Chromabond^®^, Florisil^®^, Meschery-Nagel, Düren, Germany). A MgO-SiOH-bonded silica-phase cartridge was placed in a vacuum elution apparatus and then conditioned by the successive passing of 6 mL of methanol/water, 6 mL of hexane, and 3 mL of acetonitrile. An aliquot of olive oil (2 g) mixed with 6 mL of hexane and spiked with 1 mL of syringic acid (17 µg mL^−1^) was applied to the column. The cartridge was washed twice with 3 mL of hexane. The elution conditions were optimized using a Doehlert experimental design to find the best experimental conditions of the three factors affecting the extraction process, which are: the percentage of the methanol in eluent solvent (water/methanol) (%), the volume of the eluent solvent, and the pH of the elution solvent adjusted using an acetate buffer solution: CH_3_COO-/CH_3_COOH. Finally, the extract was dried under a stream of nitrogen to 1 mL and then filtered through a 0.45 µm filter membrane for a subsequent analysis by HPLC-DAD. Recovery was calculated as follows:$$R\left(\%\right)=Y\;\left(\%\right)=100\times \frac{\left(A-A_{0}\right)}{A_{S}}$$
where A (µg mL^−1^), $A_{0}$ (µg mL^−1^), and $A_{S}$ (µg mL^−1^) are the peak areas of syringic acid determined with the spiked OO sample, non-spiked OO sample, and standard (17 µg mL^−1^), respectively.

### 3.5. Experimental Design and Statistical Analysis

The parameters of extraction were optimized using response surface methodology (RSM). Two steps are needed for multivariate optimization; the first step consists of the screening of the significant variables and response optimization using a factorial design. The second step involves the optimization of the variable response. To compare the effects of the different factors in the experimental field, concerned coded variables were used. The factors U_1_, U_2_, and U_3_ were transformed into coded variables X_1_, X_2_, and X_3_ through the following relation:$$Xi=\frac{Ui-\bar{U}i}{∆Ui}$$ where Xi is the value of the coded variable I, Ui is the value of factor I, $\bar{U}i$ is the value of factor I in the center of the experimental field, and ∆Ui is the range of variation of factor *i*.
$$\bar{U}i=\frac{Upper\;limit\;\left(Ui\right)+Lower\;limit\;\left(Ui\right)}{2}$$
$$∆Ui=\frac{Upper\;limit\;\left(Ui\right)-Lower\;limit\;\left(Ui\right)}{2}$$

The experimental response Y was represented as follows:Y = b_0_ + b_1×1_ + b_2×2_ + b_3×3_ + b_11×1_ X_1_ +b_22×2_ X_2_ + b_33×3_ X_3_ + b_12×1_ X_2_ +b_13×1_ X_3_ + b_23×2_ X_3_
where bi is the estimation of the main effects of factor I, b_ii_ is the estimation of the second-order effects, b_ij_ is the estimation of the interactions between factor I and factor j, and Y is the experimental responses (SPE Recovery %).

Therefore, many factors are likely to interact, affecting the desired response, and the only practical and fast way to optimize it is to use multivariate methods [29,40]. To optimize the SPE conditions, we used an experience design using a Doehlert matrix with N experiences (N = K_2_ + K + 1; K is equal to the parameter numbers) [29,40,56]. The appropriate conditions required for the SPE elution of phenolic compounds were determined using three important K factors: U_1_, the percentage of the methanol in the eluent solvent (water/methanol), varied between 50 and 100%; U_2_, the volume of the eluent solvent, ranged between 6 and 12 mL; and U_3_, pH of the elution solvent, comprised between 1 and 3. The maximum and minimum conditions for each variable were selected based on other studies using the same extraction procedure to analyze phenolic compounds in olive oil [57,58,59]. A constant factor is the volume of acetonitrile because it has been shown in the literature that, in the presence of this solvent, the extraction yield increases by 36%. Table 4 contains the coded values and factor levels of the Factorial and Doehlert designs applied to the elution. Design Expert Software version 13 was used for the data analysis.

## 4. Conclusions

A new method for the extraction of phenolic compounds from olive oil has been proposed. After the preliminary screening of the studied factors (methanol composition, volume of eluent, and pH), it was found that three investigated parameters showed the highest influence on the total phenolic content throughout the new-based hydrophilic interaction solid-phase extraction in olive oils. The optimum conditions were the methanol composition (90.3%), pH (2.9), and volume of eluent (7.5 mL). The yield of the total phenolics was enhanced under these conditions. The optimal conditions were applied to olive oils and showed higher concentrations of polyphenols. The advantages of this method are that it has simple extraction procedures and various phenolic compounds. This method has been considered due to its simplicity, easy handling, low cost, high efficiency, lower organic solvent consumption, and reduced extraction time. It can be used as a simple and reliable procedure in an extensive range of organic solvents for various phenolic compounds at the large-scale level and industry. Additionally, an important point for selecting this extraction method is that it should be environmentally friendly.

## Figures and Tables

**Figure 1 molecules-28-01073-f001:**
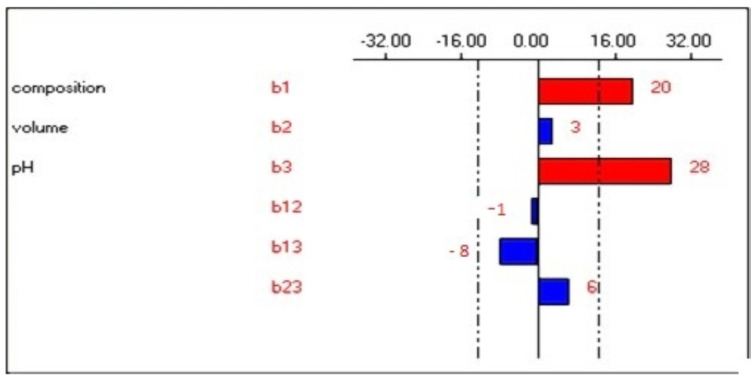
Pareto chart for the elution phase.

**Figure 2 molecules-28-01073-f002:**
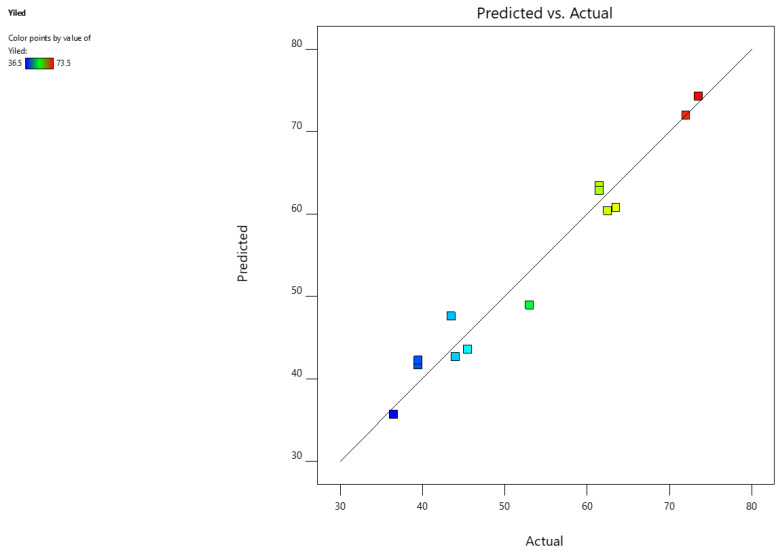
Experimental vs. predicted values for phenolic compound extraction. Y predicted: *p* < 0.00471; R^2^ adjusted = 0.96; R^2^ predicted = 0.92.

**Figure 3 molecules-28-01073-f003:**
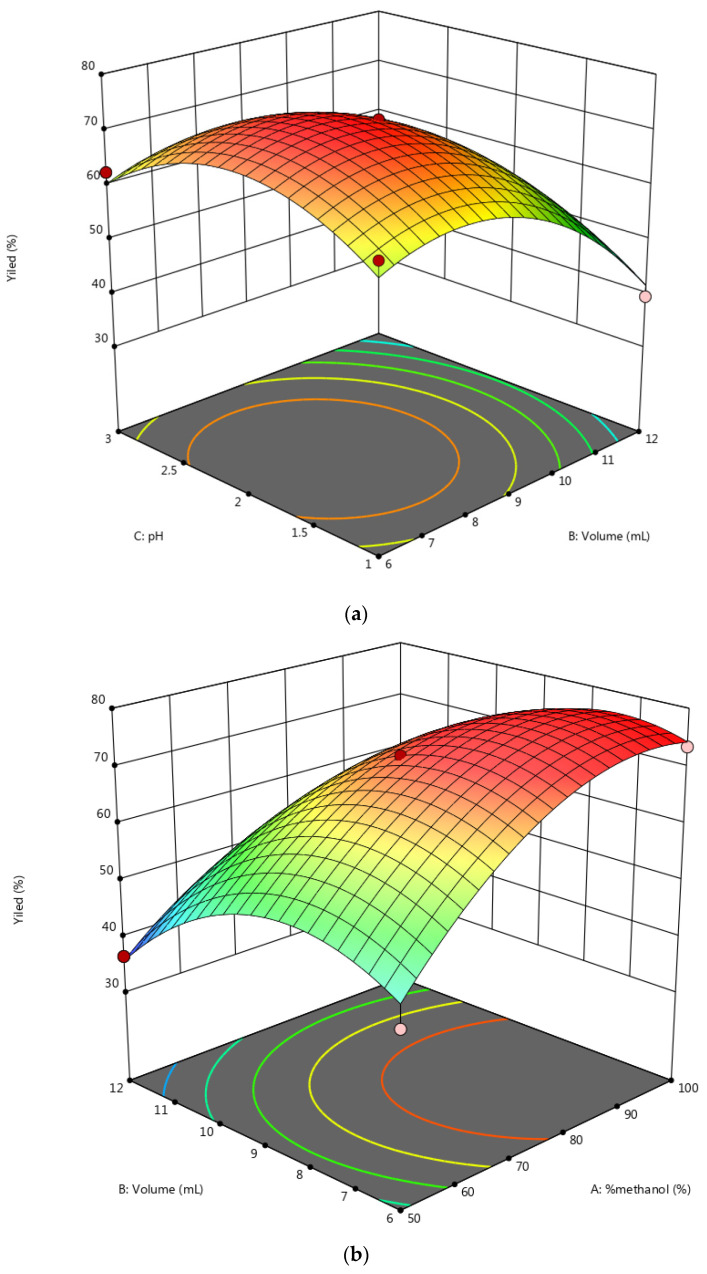
Three-dimensional (3D) response surface showing the interactions between (**a**) pH vs. volume, (**b**) volume vs. % methanol, and (**c**) pH vs. % methanol.

**Table 1 molecules-28-01073-t001:** Boehlert experimental design and the obtained responses.

N° Exp	X_1_ (%)	X_2_ (mL)	X_3_	Extraction Yield (%)
1	100	9	2	61.5
2	50	9	2	45.5
3	87.5	9	3	43.5
4	62.5	9	1	44
5	87.5	9	0.5	35
6	62.5	9	3	35
7	87.5	12	2.5	53
8	62.5	6	1.5	57.5
9	87.5	6	1.5	73.5
10	75	6	3	62.5
11	62.5	12	2.5	36.5
12	75	12	1	39.5
13	75	9	2	72

**Table 2 molecules-28-01073-t002:** Analysis of variance for the extraction yield of the phenols.

Source	Sum of Squares	df	Mean Square	F-Value	*p*-Value
Model	1917.29	9	213.03	9.20	0.0047
X_1_-Methanol	800.00	1	800.00	34.56	0.0092
X_2_-Volume	693.78	1	693.78	29.97	0.0120
X_3_-pH	0.0313	1	0.0313	0.0014	0.0043
X_1_X_2_	45.56	1	45.56	1.97	0.0025
X_1_X_3_	0.5625	1	0.5625	0.0243	0.0088
X_2_X_3_	0.2500	1	0.2500	0.0108	0.0092
X_1_^2^	195.57	1	195.57	8.45	0.0062
X_2_^2^	282.89	1	282.89	12.22	0.0039
X_3_^2^	211.75	1	211.75	9.15	0.0056
Residual	69.44	3	23.15		

**Table 3 molecules-28-01073-t003:** Phenolic compounds detected (mg/L) in different olive oil samples.

Phenolic Acids	Extra Virgin Oil Type 1	Extra Virgin Oil Type 2	Extra Virgin Oil Type 3	Olive Pomace Oil	Oil Extracted by Cold Pressing
Gallic acid	0.51	-	7.5	0.5	-
Gentisic acid	12.5	21	33	4	15
p-coumaric acid	-	-	0.5	-	-
Salicylic acid	0.37	56	92	-	33
Benzoic acid	17	9	83	-	8
o-coumaric acid	-	3	-	0.6	6.32
T-cinnamic acid	0.02	0.15	2	-	4.5
Total acid	30.4	89.15	218	5.1	66.82

**Table 4 molecules-28-01073-t004:** Investigated variables and their levels studied in the Factorial and Doehlert designs for the optimization of elution of the phenolic compounds.

Factor		Coded Level		
		−1	0	+1
X_1_	Percentage of Methanol (%)	50	75	100
X_2_	Volume (mL)	6	9	12
X_3_	pH	1	2	3

## Data Availability

Not applicable.

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
