# Peer review of "Application of Doehlert Experimental Design for Optimization of a New-Based Hydrophilic Interaction Solid-Phase Extraction of Phenolic Acids from Olive Oils"

_molecules, 2023, doi:10.3390/molecules28031073_

Round 1

Reviewer 1 Report

The work is valuable. But, the authors discussed the results poorly. Authors need to be extra careful about the reference list; lots of errors are there. Please also see the attached PDF file.

Line 35, 37, 86,.......  [1;2], [3,4;5] --- follow the journal's style

Line 68: SPE--- use the full form

Line 94: Chaal; Hikma...--- check the style

Line 132: oo--- use the full form

Line 196: Figure 1

Table 4: The column for 'extra virgin oil type 3' need to place after 'extra virgin oil type 2'

Reference list: check the journal's style-- year, journal's name, volume, page numbers

Author Response

All corrections were made in the text and in attached file

Reviewer 2 Report

In the manuscript entitled "Application of Doehlert experimental design for optimization 2 of a new based hydrophilic interaction solid-phase extraction of 3 phenolic acids from olive oils" the authors quantified phenolic compounds in olive oil using hydrophilic interaction based solid-phase extraction and optimized operating conditions using Boehlert’s design. In this regard, the Authors' contributions could be significant. However, in my opinion, some  issues must be directly addressed:

·         In the introduction section, the authors have mentioned that “Several methods have been used for the separation and identification of phenolic acids. Gas chromatography (GC) has been used 50 relatively little in the analysis of these acids.” It is necessary to include the other methods already in use for the separation and identification of phenolic acids from vegetable oil with limitations associated with them so that the scope of the current study becomes more significant.

·         In the first paragraph, the authors are emphasizing the benefits of antioxidants, while the topic of study is related to phenolic acids, it would be better to include few lines regarding “how phenols exert antioxidant effects” by citing recent articles. 

·         The SPE abbreviation must be written in full form in the introduction section at its first appearance, i.e. at the start of 3rd paragraph of the introduction.

·         A whole paragraph must be dedicated to the objective and hypothesis of the current study.

·         The sections of the manuscript should be arranged according to the journal’s guidelines.

·         The discussion part is missing from the manuscript, the results must be briefly discussed by the authors.

·         The conclusion section must incorporate the limitations of the study with a future perspective and overall possible contributions of outcomes of the study.

·         The references cited in the text and bibliography are not in accordance with the journal’s guidelines and should be thoroughly revised.

·         The manuscript must be proofread as there are numerous grammatical mistakes in the manuscript which must be corrected. 

Author Response

corrections were made in the text and in attached file in details

Reviewer 3 Report

Please separate each unit from its respective quantity value, e.g. 3% should read 3 %.

Define "OO" (Olive Oil, I guess) in section 2.4.

Section 3.1: Screening designs are used to identify, among many influencing factors, which ones are really important, i.e. have a significant effect on a particular response. I would expect more factors to be used in this design in order to demonstrate that the ones really important are the 3 selected for the optimization. If, from your experience, have identified the 3 factors I see no reson to use a screening design.

3.2.1: "to estimate the importance of each factor alone" better to say "to estimate the main effect of each influencing factor".

Table 1 shows that pH was varied from 1-3. Table 2 shows that pH was, effectively, varied from 0.5-3. Is 0.5 a mistake? Please review. Figure 3 confirm that pH varies from 1-3.

Author Response

All coorrection were in the text and in attached file
